
# Casting a graph net to catch dark showers

**Elias Bernreuther**[⋆], **Thorben Finke**[†], **Felix Kahlhoefer**[‡],
**Michael Krämer**[◦] **and Alexander Mück**[×]

Institute for Theoretical Particle Physics and Cosmology (TTK),
RWTH Aachen University, D-52056 Aachen, Germany

⋆ ebernreuther@physik.rwth-aachen.de, † finke@physik.rwth-aachen.de,
‡ kahlhoefer@physik.rwth-aachen.de, ◦ mkraemer@physik.rwth-aachen.de,
× mueck@physik.rwth-aachen.de

## Abstract

Strongly interacting dark sectors predict novel LHC signatures such as semi-visible jets resulting from dark showers that contain both stable and unstable dark mesons. Distinguishing such semi-visible jets from large QCD backgrounds is difficult and constitutes an exciting challenge for jet classification. In this article we explore the potential of supervised deep neural networks to identify semi-visible jets. We show that dynamic graph convolutional neural networks operating on so-called particle clouds outperform convolutional neural networks analysing jet images as well as other neural networks based on Lorentz vectors. We investigate how the performance depends on the properties of the dark shower and discuss training on mixed samples as a strategy to reduce model dependence. By modifying an existing mono-jet analysis we show that LHC sensitivity to dark sectors can be enhanced by more than an order of magnitude by using the dynamic graph network as a dark shower tagger.



# 1 Introduction

The huge wealth of data taken at the LHC offers a unique opportunity to explore the properties of dark sectors and uncover the nature of dark matter (DM). At the same time, such an amount of data poses an unprecedented challenge to precisely determine and efficiently suppress backgrounds in order to identify potential signals of new physics. As the complexity of experimental analyses increases, there has been rapidly growing interest in using machine learning techniques to distinguish signal from background. For example, deep neural networks are powerful tools for the classification of jets, which can significantly improve the sensitivity to new physics signals hidden in QCD background, see e.g. the reviews [1, 2]. A particularly interesting and well-motivated case are so-called dark showers [3–11], which may resemble QCD jets even though they result from new interactions and contain exotic particles.

Dark showers arise in extensions of the Standard Model (SM) that contain new strong dynamics, i.e. exotic fermions charged under a new gauge group that confines at low energies [8–33]. If these fermions are produced at the LHC (for example via a heavy mediator), they will undergo fragmentation and hadronisation similar to SM quarks. These processes then lead to a shower of composite states that are neutral under the new gauge group, which in analogy to the SM we will refer to as dark mesons and dark baryons. The detailed spectrum of such a dark sector is difficult to predict from first principles, but a common feature of many models is the existence of dark pions $\pi_d$, which are the Pseudo-Goldstone bosons from the spontaneous breaking of chiral symmetry, and dark vector mesons $\rho_d$, which have a mass similar to the confinement scale $\Lambda_d$.

What makes dark showers exciting to study at the LHC is the fact that some of the dark mesons (e.g. the dark pions) may be stable on cosmological scales, thus providing a potential explanation for DM, while other dark mesons (e.g. the dark vector mesons) may decay on collider scales into SM particles, and in particular into SM quarks. This combination of visible and invisible particles in the same shower then leads to so-called semi-visible jets [4,5,7,9–11]. The fraction of invisible particles in a dark shower will fluctuate around the expectation value $r_{inv}$, leading to sizeable amounts of missing energy even if two dark showers are produced back-to-back. Moreover, there is a finite probability that a dark shower will remain completely invisible, in which case the resulting signature is a mono-jet signature with a single semi-visible jet pointing in the direction opposite to the missing energy vector.

Although LHC searches for jets and missing energy are not optimised for semi-visible jets, their sensitivity can be enhanced significantly by suppressing QCD backgrounds with improved jet classification. Traditional jet tagging algorithms rely on hand-crafted high-level features such as $N$-subjettiness [34]. Basic machine learning algorithms like boosted decision trees can then learn decision boundaries along those high-level features for classification. Deep learning algorithms on the other hand are able to work on low-level quantities, such as particle four-momenta, and extract complex features relevant for the classification. For an overview of deep learning jet taggers and their performance on separating hadronic top jets from light QCD jets we refer to ref. [35].

In this article we explore the potential of supervised deep neural networks to identify semi-visible jets. For this purpose we consider three different architectures: a convolutional neural network (CNN) working on jet images [36], a Lorentz layer (LoLa) network [37] working on an ordered set of four-momenta of jet constituents and the dynamic graph convolutional neural network (DGCNN) [38] of ref. [39] working on an unordered set of particles, a so-called particle cloud, similar to the concept of point clouds in computer vision.[1] While the different techniques show similar performance on the task of top classification, we show that their performances differ more significantly for the classification of semi-visible jets. In particular

---

[1]For other examples of graph networks in the context of LHC physics see e.g. refs. [40–45].

dynamic graph neural networks are shown to be powerful tools for tagging semi-visible jets. We find that it may be possible to enhance the sensitivity of LHC searches for signatures with semi-visible jets by an order of magnitude through jet classification with a DGCNN.

This paper is structured as follows. In section 2 we discuss the properties of dark showers and in particular semi-visible jet signatures. The neural networks used in our analysis are introduced in section 3. We compare the classification performance of the different neural network architectures and study the dependence of the jet identification on the parameters of the dark sector model. To demonstrate the potential of a semi-visible jet classifier, we adapt an existing mono-jet search to use a jet classifier based on a dynamic graph neural network in section 4. Our conclusions are presented in section 5. In appendix A we describe the generation of signal and background events or jets. The architectures of the neural networks employed in our analysis are presented in detail in appendix B.

## 2  Dark sector models and semi-visible jets

The structure of a semi-visible jet is mostly characterised by three parameters: The fraction of invisible particles $r_{\text{inv}}$, the mass $m_{\text{meson}}$ of the unstable dark mesons and the confinement scale $\Lambda_{\text{d}}$. Indeed, even for $r_{\text{inv}} = 0$ dark showers may differ substantially from ordinary QCD jets, because of the different running of the dark gauge coupling, the absence of heavy quarks in the shower and the presence of substructure corresponding to the decays of individual dark mesons. With increasing $r_{\text{inv}}$ the semi-visible jet becomes increasingly different from QCD jets, but also harder to study because of the smaller number of visible constituents.

In the following we will investigate how the properties of semi-visible jets enable us to distinguish them from SM backgrounds. Unless explicitly stated otherwise, we will assume that the dark mesons have a mass close to the confinement scale: $m_{\text{meson}} \approx \Lambda_{\text{d}}$. Furthermore, we will limit ourselves to the case of a dark $SU(3)$ gauge group with two flavours of dark quarks $q_{\text{d}}$. This choice is based on a recent study on strongly interacting dark sectors [10], which identified this scenario as particularly interesting, because it allows for a viable DM candidate consistent with all cosmological and laboratory constraints. In the following we briefly review the key aspects of this model, which will serve as a benchmark scenario for the present study.

The two dark quarks $q_{\text{d}}$ are assumed to be in the fundamental representation of the dark $SU(3)$ gauge group and carry opposite charges with respect to an additional $U(1)'$ gauge symmetry. Below the confinement scale the dark quarks form mesons, which we denote by $\pi_{\text{d}}^0, \pi_{\text{d}}^\pm$ and $\rho_{\text{d}}^0, \rho_{\text{d}}^\pm$ in analogy to their QCD counterparts, with the superscripts denoting the $U(1)'$ charges. The dark sector is therefore characterised by the meson masses $m_{\pi_{\text{d}}}$ and $m_{\rho_{\text{d}}}$ and the coupling strength $g_{\text{d}}$ of their interactions. We furthermore assume that the dark baryons are sufficiently heavy that they are irrelevant for phenomenology. It can be shown that in this set-up all dark pions are stable on cosmological scales and therefore constitute a potential DM candidate.

The interactions of the dark sector with the SM are mediated by the massive $U(1)'$ gauge boson $Z'$ with vector couplings to both dark and SM quarks, denoted $e_{\text{d}}$ and $g_q$, respectively. Couplings to leptons, as well as mixing between the $Z'$ and SM gauge bosons, are assumed to be suppressed. In analogy to $\gamma$-$\rho^0$ mixing in the SM, the $Z'$ mixes with the $\rho_{\text{d}}^0$, which induces small couplings between the $\rho_{\text{d}}^0$ and SM quarks and renders the $\rho_{\text{d}}^0$ unstable. For $m_{\rho_{\text{d}}} < 2m_{\pi_{\text{d}}}$ the $\rho_{\text{d}}^\pm$ mesons can only decay into three-body final states via an off-shell $Z'$, which makes them stable with respect to collider phenomenology. We assume that each mesonic degree of freedom is produced with the same probability during the dark hadronisation process while the production of dark baryons in the shower is negligible, and that the $\rho_{\text{d}}^0$ mesons decay

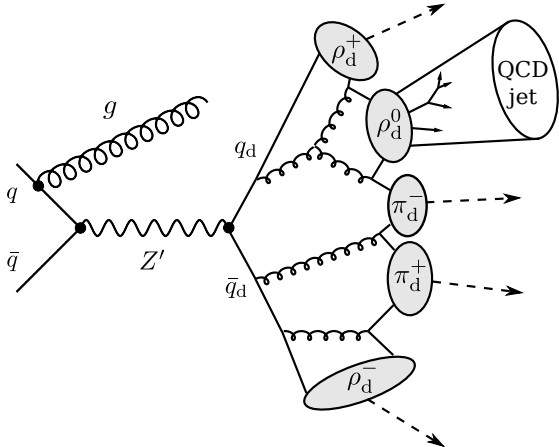

Figure 1: Schematic illustration of a dark shower from the decay of a $Z'$ produced in association with a gluon. Figure taken from ref. [10].

promptly.[2] The invisible energy fraction in a dark shower is then given by $r_{\text{inv}} = 0.75$, which we will use as the benchmark value in the following. Furthermore, the relevant mass for characterising the dark shower is the mass of the dark vector mesons: $m_{\text{meson}} = m_{\rho_{\text{d}}}$.

We note in passing that the assumption $m_{\rho_{\text{d}}} < 2m_{\pi_{\text{d}}}$ can be motivated from cosmology, because the relic density of dark pions is determined by the rate of the annihilation process $\pi_{\text{d}} \pi_{\text{d}} \to \rho_{\text{d}} \rho_{\text{d}}$, which becomes Boltzmann suppressed at low temperatures. Provided $m_{\pi_{\text{d}}}$ and $m_{\rho_{\text{d}}}$ are sufficiently close, the observed relic abundance can be reproduced even for weak portal interactions and/or heavy $Z'$ bosons, which makes it possible to satisfy constraints from direct detection experiments. For example, for $m_{\pi_{\text{d}}} = 4$ GeV and $g_{\text{d}} = 1$ one requires $m_{\rho_{\text{d}}} \approx 5$ GeV, while the $Z'$ mediator can be in the TeV range [10].

LHC phenomenology for this model is then dominated by the on-shell production of the $Z'$ (possibly in association with SM particles) and its subsequent decays into either SM or dark quarks. While the former case leads to di-jet resonances that can be easily reconstructed, the latter case gives rise to more challenging semi-visible jets, see figure 1. Although existing LHC searches for missing energy have some sensitivity to this set-up, they are not optimised for the case of dark showers, where the missing energy tends to be aligned with a visible jet. The reason is that such a configuration is difficult to disentangle from QCD backgrounds resulting from mis-reconstructed jets [46,47]. A detailed reinterpretation of existing exclusion limits from a search for di-jet resonances [48] and searches for missing energy [46, 49, 50] was performed in ref. [10]. It was found that for the benchmark values $m_{q_{\text{d}}} = 500$ MeV, $m_{\pi_{\text{d}}} = 4$ GeV, $m_{\rho_{\text{d}}} = \Lambda_{\text{d}} = 5$ GeV and $m_{Z'} = 1$ TeV couplings of the order of 0.1 are still consistent with all constraints, even though the production cross section for dark showers is of the order of picobarn.

In order to enhance experimental sensitivity to dark showers it is essential to improve background suppression, which potentially allows for other selection cuts to be relaxed. The most promising strategy for doing so is to develop techniques for distinguishing semi-visible jets from QCD jets and extend existing analyses by a dedicated tagger for semi-visible jets. In the following section we will study how to achieve this goal with a neural network trained to identify dark showers.

---

[2]We note that for small $Z'$ couplings the $\rho_{\text{d}}^0$ can be long-lived and lead to displaced vertices at the LHC. The corresponding production cross sections can nevertheless be sufficiently large that thousands of such events have already gone unnoticed at ATLAS and CMS. Ongoing detector upgrades as well as new analysis strategies make these signatures a promising target for future LHC runs. Exploring the sensitivity of searches for displaced vertices for dark sector models is subject of separate work in progress.

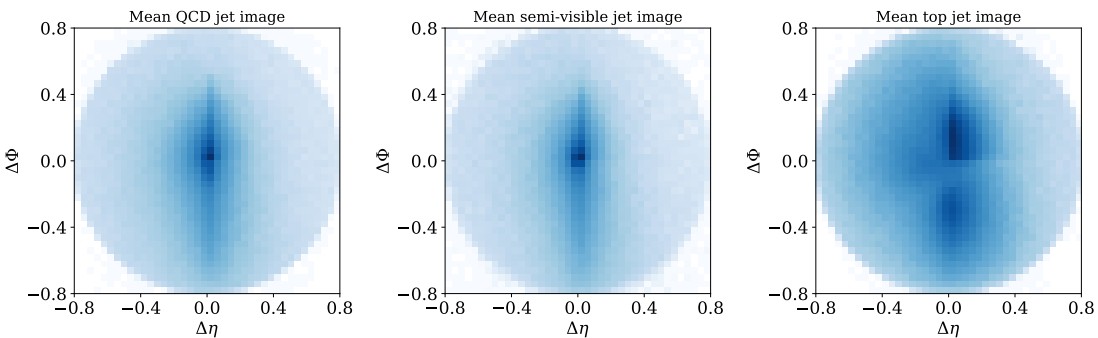

Figure 2: Average jet images in the plane of pseudo-rapidity $\eta$ and azimuthal angle $\phi$ for light QCD (left), semi-visible dark jets (middle), and boosted top jets (right). See appendices A and B for details on the event generation and the preprocessing steps for the generation of the images.

## 3 Deep neural networks for semi-visible jets

Deep neural networks have been applied to a wide range of jet classification problems, including the identification of jet substructures. It is not clear a priori how the information encoded in a jet should be mapped to a particular data structure. The representation of jets as images is motivated by jet reconstruction with calorimeters [51–53]. Convolutional neural networks (CNNs) are a powerful tool to analyse jet images. CNNs apply convolution filters that operate on small windows of an image array and allow for an efficient identification of features in the image. Convolutional networks have been very successful in the identification of jet substructures, for example in the case of top jets, and serve as a benchmark for assessing jet classification tools [35, 36, 54, 55].

To illustrate the challenge of identifying semi-visible jets based on images, we show average jet images for light QCD di-jets, semi-visible jets and hadronically decaying boosted top jets in figure 2. The event generation and the preprocessing steps for the generation of the images are described in appendices A and B, respectively. The average top jet has a clearly visible substructure originating from the hadronic top decay and thus differs substantially from light QCD jets. The semi-visible jets from the dark shower, on the other hand, are very similar to QCD.

Instead of an image, a jet may be represented as a collection of particle constituents. A strong top-jet identification performance can be achieved with so-called Lorentz-layer (LoLa) networks. These architectures map constituent four-vectors to quantities more directly related to physical observables, such as invariant masses, transverse momenta or linear combinations of constituent energies [37, 56–58]. The input to the Lorentz layer typically consists of the original particle four-momenta complemented by various learned linear combinations of those. Providing learned linear combinations allows the network to identify jet substructures more efficiently. The constituent four-momenta together with the learned linear combinations are then transformed into invariant masses and other physical observables by the Lorentz layer, before classification by a fully connected neural network.

Dynamic graph convolutional neural networks (DGCNNs) [38] are another class of powerful classifiers which apply so-called edge convolutions to particle constituents, or particle clouds, characterised by features such as energies, transverse momenta, or angular separations [39]. The edge convolution differs from a convolution over an image in the definition of the local patch that the convolution kernel observes. In an image, the local patch corresponds to some neighbourhood of pixels. For an edge convolution, a local graph is constructed for

each point in the particle cloud from its nearest neighbours using a distance measure in the space of input features. Calculating new nearest neighbours dynamically from the output of the previous edge convolution allows for particles that are initially far apart to become close in feature space already for the next convolution. In this way, long range correlations can be accessed efficiently with few convolutional layers and the network is potentially able to learn the graph structure that offers most information. For a cloud of particles representing a jet, it appears natural that the correlation of particles which are not close in the initial features, can be important for the classification of the jet. The dynamic update enables the network to link those initially distant particles.

In the following, we will analyse and compare the classification performance of a CNN, a LoLa network and a DGCNN for semi-visible jets against light QCD background jets. For comparison, we also show results for the well-established benchmark task of top-jet identification. The architectures of the CNN, the LoLa network and the DGCNN are described in detail in appendix B.

## 3.1 Classification performance

Our neural networks are trained on 200k background and 200k signal jets, with a validation split of 10%. For the signal generation we use the benchmark parameters introduced above, i.e. $r_{\mathrm{inv}} = 0.75$, $m_{\mathrm{meson}} = 5\,\mathrm{GeV}$ and $m_{Z'} = 1\,\mathrm{TeV}$. The network performance results presented below are based on an independent test set of 100k background and 100k signal jets. Note that for the moment we use the same dark sector parameters for training and testing, even though these parameters would be unknown in a realistic setting. We will return to the issue of model dependence in section 3.2 and present a mitigation strategy in section 3.3.

The networks output two numbers for each jet, which can be interpreted as the probabilities to belong to the background or the signal class, respectively. Defining a threshold probability necessary for a jet to be labelled as signal and scanning over this threshold, one obtains the receiver operating characteristic (ROC) curve, i.e. the inverse of the fraction of background jets passing the threshold (the background rejection $1/\epsilon_B$) as a function of the fraction of signal events passing the threshold (the signal efficiency $\epsilon_S$). Figure 3 shows the ROC curve for semi-visible jet identification (left panel) and for top-jet identification (right panel). To estimate the stability and reproducibility of the network performances, five networks with independent random weight initialisations are trained on the same training data and tested on the same testing data. The small spread in performance indicated by the shaded band around the ROC curves in figure 3 shows that the training convergence of the networks is stable. For a comparison of the top-tagging performance of our networks with the results of ref. [35], see figure 10 in appendix B.

Various network performance measures are collected in table 1. We display the accuracy, i.e. the ratio of the number of correctly classified jets over the total number of jets, the area under the ROC curve, $\mathrm{AUC} = \int \mathrm{d}\epsilon_B\, \epsilon_S(\epsilon_B)$, and the background rejection at a signal efficiency of 30%. Error estimates correspond to the spread obtained from the five independent network trainings mentioned above.

The results presented in figure 3 and table 1 first of all confirm that the classification of semi-visible jets is more challenging than that of top jets. Comparing the CNN, LoLa and DGCNN architectures, we find that the DGCNN performs best for both top and semi-visible jet identification. While the difference between the CNN, the LoLa network and the DGCNN is moderate for top identification, the strength of the DGCNN is particularly significant for the classification of semi-visible jets. As shown in figure 3, the background rejection at a given signal efficiency, which is most relevant for an experimental analysis, is significantly improved by a DGCNN for a wide range of signal efficiencies. Specifically, at a signal efficiency of 30%, the background rejection of the DGCNN is almost a factor of five stronger than that of the

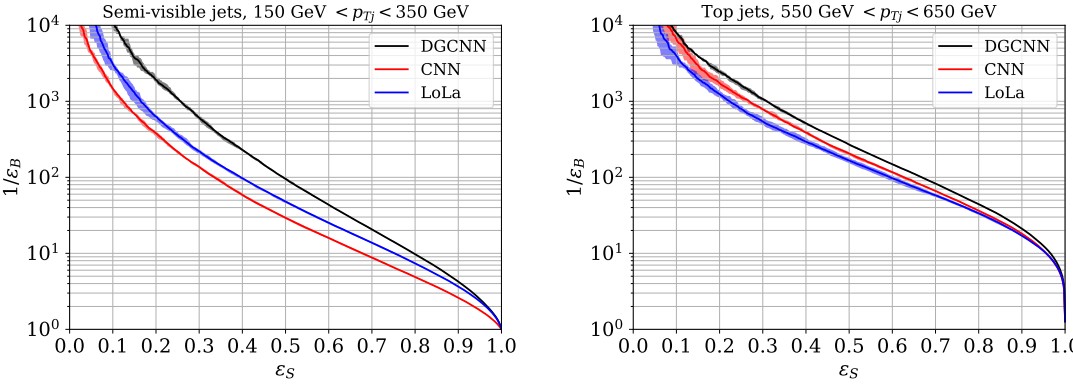

Figure 3: Comparison of the ROC curves in background rejection $1/\epsilon_B$ and signal efficiency $\epsilon_S$ for semi-visible jet identification (left panel) and for boosted top jet identification (right panel) as obtained by the CNN, LoLa and DGCNN architectures, respectively. The error bands correspond to the spread obtained from five independent initialisations of the network.

CNN.

## 3.2 Model dependence of semi-visible jet classification

In this section we explore the model dependence of the semi-visible jet classification with the DGCNN, i.e. we study how the performance changes as we vary the parameters of the strongly interacting dark sector. This not only sheds light on how much a specific network generalises to other dark shower scenarios, but it also provides some indication of which signal features the network may learn.

As a crucial parameter, the invisible fraction $r_{\text{inv}}$ represents the average percentage of missing energy and characterises the composition of the dark showers. The model described in section 2 predicts $r_{\text{inv}} = 0.75$. However, for the purpose of this section we treat $r_{\text{inv}}$ as a phenomenological parameter which can assume any value between zero and one. To this end, we decay all the dark mesons in PYTHIA with branching ratio $r_{\text{inv}}$ into invisible particles and branching ratio $1 - r_{\text{inv}}$ into Standard Model quarks, respectively. Training and testing the DGCNN classifier architecture on dark shower samples with different values of $r_{\text{inv}}$ we obtain the ROC curves shown in the left panel of figure 4. We find that dark showers with larger $r_{\text{inv}}$ are in general easier to distinguish from QCD. For $0.1 < \epsilon_S < 0.3$, which is the most interesting range for improving an analysis with the jet tagger, the background suppression varies by roughly an order of magnitude. Note that for very small $r_{\text{inv}}$ the classification performance increases again as almost all the energy from the $Z'$ resonance ends up in visible jets leading to a harder jet $p_T$ distribution which is more different from QCD.

As another dark sector parameter we vary the dark meson mass $m_{\text{meson}} = m_{\pi_d} = m_{\rho_d}$, together with the dark confinement scale $\Lambda_d = m_{\text{meson}}$. Note that the small mass splitting between the $\pi_d$ and $\rho_d$ motivated by cosmology has no impact on the LHC phenomenology. Larger values of $\Lambda_d$ lead to a stronger running of the dark sector coupling $\alpha_d$ at the energy scale of the semi-visible jet. Among other effects, the running of $\alpha_d$ changes jet substructure observables such as the distribution of the two-point energy correlation function discussed in ref. [11]. Moreover, as the jet constituents arise from dark meson decays they encode the dark meson mass scale $m_{\text{meson}}$. As shown in the right panel of figure 4, changing the confinement and meson mass scale between 5 GeV and 20 GeV has no significant effect on the classification performance.

Table 1: Performance measures for classifying semi-visible jets and top jets by the CNN, LoLa and DGCNN architectures, respectively, corresponding to the ROC curves presented in figure 3. We show the accuracy, the AUC value and the background rejection at a signal efficiency of 30%. The central value is the mean of the five independent training runs of the network, while the error estimate corresponds to the spread in the performance.

|  |  | Acc [%] | AUC | $1/\epsilon_B$ ($\epsilon_S = 0.3$) |
|---|---|---|---|---|
| semi-visible jets | CNN | $79.88^{+0.21}_{-0.22}$ | $0.8790^{+0.0019}_{-0.0019}$ | $137^{+6}_{-4}$ |
|  | LoLa | $83.26^{+0.14}_{-0.13}$ | $0.9118^{+0.0008}_{-0.0010}$ | $220^{+11}_{-17}$ |
|  | DGCNN | $85.04^{+0.12}_{-0.08}$ | $0.9258^{+0.0007}_{-0.0007}$ | $608^{+36}_{-40}$ |
| top jets | CNN | $92.98^{+0.05}_{-0.09}$ | $0.9802^{+0.0002}_{-0.0005}$ | $785^{+40}_{-29}$ |
|  | LoLa | $92.83^{+0.11}_{-0.11}$ | $0.9791^{+0.0007}_{-0.0008}$ | $540^{+77}_{-53}$ |
|  | DGCNN | $93.47^{+0.06}_{-0.06}$ | $0.9831^{+0.0001}_{-0.0002}$ | $1073^{+47}_{-75}$ |

Since the values of the dark sector parameters are not known a priori, it is a relevant question to which extent the classification is model-dependent. Therefore, we next consider samples with different dark sector parameters for training and testing. We train the network on dark showers with our benchmark parameters $r_{\mathrm{inv}} = 0.75$ and $m_{\mathrm{meson}} = 5$ GeV, and evaluate the performance on a range of samples with different choices for $r_{\mathrm{inv}}$ and $m_{\mathrm{meson}}$, respectively. The corresponding ROC curves are displayed in figure 5. ROC curves for training and testing with identical parameter values (see figure 4) are shown for comparison. As expected, we find a drop in performance as the difference between the model parameter settings in the training and test samples increases. Varying $r_{\mathrm{inv}}$, the decrease in performance is modest. Only for $r_{\mathrm{inv}} = 0.1$, the drop is larger since the tagger cannot benefit any more from the harder jet $p_T$ distribution. The model dependence is significantly more pronounced for the dark meson mass. The background suppression is reduced by about an order of magnitude for $0.1 < \epsilon_S < 0.3$, indicating that the network learns to reconstruct the dark meson mass to some extent from the constituents.

### 3.3 Mitigating model dependence with mixed samples

A simple way to mitigate this behaviour and provide a more model-independent semi-visible jet classifier, is to train the network on a mixed sample, which contains a range of different $r_{\mathrm{inv}}$ values or dark meson masses. Here we consider a mixed $r_{\mathrm{inv}}$ sample containing an equal number of jets with $r_{\mathrm{inv}} = 0.1, 0.5, 0.75$, and $0.9$, as well as a mixed meson mass sample consisting of an equal number of jets with $m_{\mathrm{meson}} = 5$ GeV, 10 GeV, and 20 GeV. This way the network is forced to learn features common to the different samples instead of learning to reconstruct, for example, one specific dark meson mass. The performance of such a more general classifier is significantly better than that of a classifier trained on specific values of $r_{\mathrm{inv}}$ and $m_{\mathrm{meson}}$ when both are applied to a wider range of model parameters, see figure 6 and table 2.[3] A significant improvement is also present for dark meson masses that were not included in the mixed training sample, as the results for $m_{\mathrm{meson}} = 15$ GeV show. Note that it may be possible to use

---

[3]We note that the DGCNN significantly outperforms both the CNN and the LoLa architecture also for training on mixed samples.

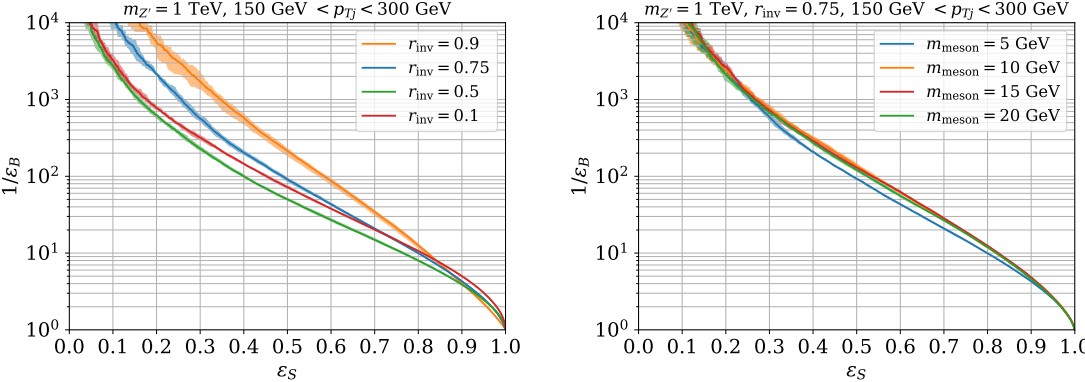

Figure 4: DGCNN ROC curves for the discrimination of dark showers from QCD for different values of $r_{\mathrm{inv}}$ (left panel) and $m_{\mathrm{meson}}$ (right panel). The error bands correspond to the spread obtained from five independent initialisations of the network.

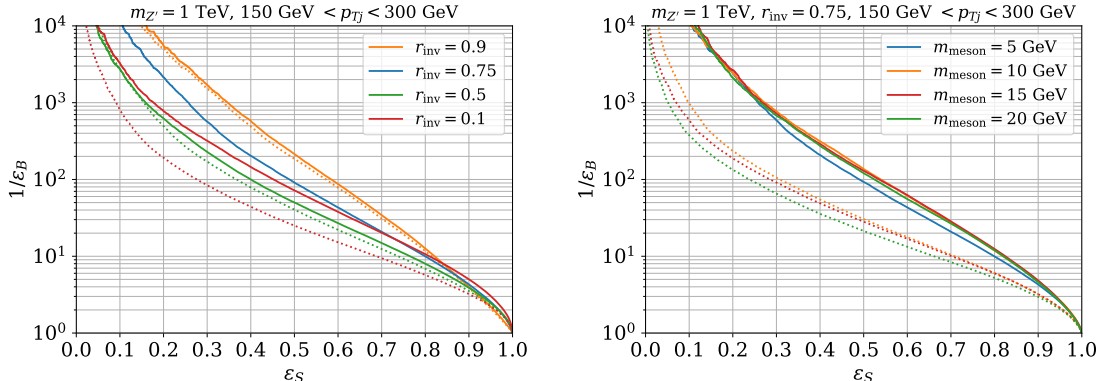

Figure 5: ROC curves (dotted lines) for the DGCNN trained on dark showers with the benchmark values ($r_{\mathrm{inv}} = 0.75$ and $m_{\mathrm{meson}} = \Lambda_{\mathrm{d}} = 5$ GeV) and tested on different values of $r_{\mathrm{inv}}$ (left panel) and $m_{\mathrm{meson}}$ (right panel). ROC curves for training and testing on samples with identical parameters are shown for comparison (solid lines, as in figure 4).

networks that each have learned a specific dark meson mass to reconstruct this mass from a possible dark shower signal, e.g. with a parametrised network [59].

We have also studied the dependence of the network training and performance on the $Z'$ mediator mass. We find only small differences in the ROC curves when varying the $Z'$ mass between 1 TeV and 2 TeV. Moreover, training the network on a $Z'$ mass different from the mass used in the test sample only has a small effect on the network performance.

## 4   Mono-jet analysis with machine learning

In this section, we study the sensitivity improvement for a dark shower search with the help of a DGCNN classifier. As an example we consider the ATLAS mono-jet analysis with 36.1 fb$^{-1}$ [46] applied to a dark shower signal with the benchmark parameters from ref. [10], i.e. $r_{\mathrm{inv}} = 0.75$, $m_{\pi_{\mathrm{d}}} = 4$ GeV and $m_{\rho_{\mathrm{d}}} = \Lambda_{\mathrm{d}} = 5$ GeV. The mono-jet search is sensitive to dark shower events

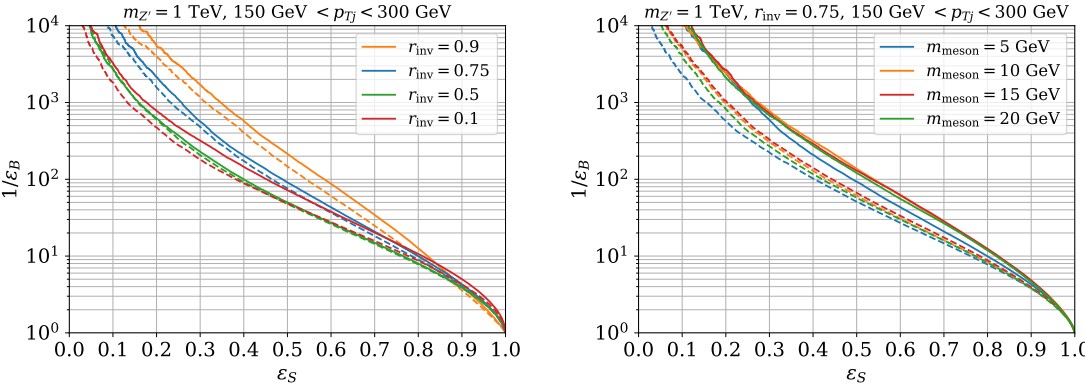

Figure 6: ROC curves (dashed lines) for the DGCNN trained on mixed samples of dark showers with different values of $r_{\mathrm{inv}}$ and $m_{\mathrm{meson}}$, and tested on pure samples each containing a specific value for $r_{\mathrm{inv}}$ (left panel) and $m_{\mathrm{meson}}$ (right panel). ROC curves for training and testing on samples with identical parameters are shown for comparison (solid lines, as in figure 4).

where one of the dark showers stays invisible leading to a mono-jet topology, i.e. a large angular distance $\Delta\phi \approx \pi$ between missing energy and the semi-visible jet.[4] To identify these jets as originating from a dark shower, we integrate our graph network into the analysis as a dark shower tagger.

To generate signal and background events, we use the tools described in appendix A. Note that we focus on the dominant $Z$+jets background. The signal jets for training the tagger are extracted from a dark shower signal with $m_{Z'} = 1$ TeV. The sample of a given signal region consists of all fat jets from the corresponding signal events, where we require a truth-level dark quark within the jet cone. We emphasise that we apply two different jet definitions to each event. While the signal events are defined using the ATLAS jet definition of ref. [46], the fat jets for the tagger training are anti-$k_T$ jets with a minimal transverse momentum of 100 GeV and a jet radius $R = 0.8$ in order to contain all radiation from an underlying dark quark. The background jet samples consist of all fat jets from the corresponding $Z$+jets events. We train on 200k signal and 200k background jets.

In the analysis we first apply the cuts from ref. [46]. We then sort the remaining events into signal regions and apply the DGCNN dark shower tagger, trained on the appropriate signal region, to all fat jets in each event. If at least one of the jets in an event is tagged as a dark shower jet the event is accepted. Otherwise the event is rejected. By varying the tagging threshold we control the signal event efficiency and $Z$+jets background rejection rate. The corresponding ROC curve is shown in the left panel of figure 7 for the signal region EM4, which corresponds to 400 GeV$< \not{E}_T < 500$ GeV. EM4 is the signal region most sensitive to the dark shower signal with our benchmark parameters. The efficiencies $\epsilon_S$ and $\epsilon_B$ shown in figure 7 are relative to the event numbers after the ordinary mono-jet cuts. Hence, the existing analysis without a dark shower tagger is equivalent to the point $\epsilon_S = \epsilon_B = 1$ in the lower right-hand corner of the plot.

To estimate the influence of detector effects on the DGCNN tagger, we also show the analogous ROC curve for a tagger based on detector level quantities, i.e. towers and tracks from DELPHES [60], instead of particles as input in the training and in the analysis. We find that detector effects lead to a slightly reduced background rejection compared to the case with

---

[4]Other event topologies, where both dark showers are visible and recoil against an ISR jet to obtain a sizeable $\Delta\phi$, have been shown to be sub-leading in ref. [10].

Table 2: Performance measures for the DGCNN for different dark meson masses. We show the performance for networks trained and tested on the same $m_{\text{meson}}$, trained on $m_{\text{meson}} = 5$ GeV, and trained on a mixed sample, corresponding to the ROC curves in the right panels of Figs. 4, 5, and 6. We show the accuracy, the AUC value and the background rejection at a signal efficiency of 30%. The central value is the mean of five independent training runs, while the error estimate corresponds to the spread in the performance.

| $m_{\text{meson}}$ [GeV] test | $m_{\text{meson}}$ [GeV] training | Acc [%] | AUC | $1/\epsilon_B$ ($\epsilon_S = 0.3$) |
|---|---|---|---|---|
| 5 | 5 | $85.14^{+0.04}_{-0.06}$ | $0.9267^{+0.0002}_{-0.0005}$ | $589^{+47}_{-46}$ |
| | mixed | $83.61^{+0.09}_{-0.09}$ | $0.9148^{+0.0009}_{-0.0008}$ | $224^{+20}_{-15}$ |
| 10 | 10 | $86.04^{+0.05}_{-0.05}$ | $0.9333^{+0.0004}_{-0.0004}$ | $774^{+67}_{-59}$ |
| | 5 | $81.2^{+0.3}_{-0.2}$ | $0.8965^{+0.0015}_{-0.0012}$ | $106^{+12}_{-6}$ |
| | mixed | $84.03^{+0.05}_{-0.03}$ | $0.9180^{+0.0004}_{-0.0003}$ | $304^{+6}_{-7}$ |
| 15 | 15 | $86.24^{+0.03}_{-0.03}$ | $0.9336^{+0.0002}_{-0.0002}$ | $720^{+43}_{-53}$ |
| | 5 | $81.00^{+0.17}_{-0.18}$ | $0.8950^{+0.0005}_{-0.0012}$ | $91^{+6}_{-3}$ |
| | mixed | $84.38^{+0.11}_{-0.12}$ | $0.9198^{+0.0007}_{-0.0007}$ | $330^{+16}_{-15}$ |
| 20 | 20 | $86.03^{+0.09}_{-0.06}$ | $0.9316^{+0.0006}_{-0.0004}$ | $682^{+43}_{-33}$ |
| | 5 | $79.2^{+0.2}_{-0.3}$ | $0.883^{+0.001}_{-0.002}$ | $65^{+2}_{-2}$ |
| | mixed | $83.96^{+0.08}_{-0.08}$ | $0.9161^{+0.0011}_{-0.0009}$ | $270^{+15}_{-16}$ |

particles as DGCNN input.

Using the improved background suppression due to the DGCNN tagger in the mono-jet search, we derive an expected limit on the dark shower cross section. The background event numbers $B$ and systematic uncertainties $\Delta B$ from the ATLAS analysis [46] are divided by the background rejection obtained from our simulation of the dominant $Z$+jets background. We apply the same additional rejection rate for the sub-leading background of $W$+jets events. Furthermore we assume that contributions from other backgrounds, in particular from di-bosons as well as $t\bar{t}$ and single tops, are still negligible in the analysis with the tagger. This assumption is based on the fact that dark showers are easier to distinguish from top jets than from QCD jets, which should result in a tagger rejection rate of the top background that is at least comparable to the rate for the $V$+jets background. Moreover these backgrounds would still have little bearing on the final limits even if the rejection were significantly worse than for $V$+jets. Hence, we apply the same universal rejection factor to all background contributions and their respective systematic uncertainties. We derive the expected 95% C.L. limit on the number of signal events assuming that the number of observed events is equal to the background prediction. Hence, we construct the profile likelihood [61]

$$\mathcal{L}(\mu) = \frac{1}{B!}\left(\mu S + B\left(1 + \frac{\Delta B}{B}\theta_B\right)\right)^B e^{-\left(\mu S + B\left(1 + \frac{\Delta B}{B}\theta_B\right)\right)} e^{-\theta_B^2/2}, \tag{1}$$

with the value of the nuisance parameter $\theta_B$ chosen such that it maximises the likelihood for a given signal strength $\mu$. We obtain the limit by excluding points for which the log-likelihood

ratio

$$q_\mu = -2 \left( \log \mathcal{L}(\mu = 1) - \log \mathcal{L}(\mu = 0) \right) \tag{2}$$

is larger than 3.84, which corresponds to a $p$-value of 0.05 for a $\chi^2$ distribution with 1 degree of freedom. Thus we arrive at a limit on the number of signal events for a given signal region.

If backgrounds can be estimated directly from data in the same way as for existing monojet searches (e.g. using $Z(\to \mu\mu) + \text{jets}$ as a control region for $Z(\to \nu\nu) + \text{jets}$), using a dark shower tagger should not significantly increase the relative systematic uncertainties of the background estimates (apart from decreasing the number of events in the control region, hence increasing the corresponding statistical uncertainty). As indicated by the background numbers and uncertainties in table 3, applying the dark shower tagger then takes the search from a systematics dominated to a strongly statistics dominated regime. In other words, the number of signal events that the search is sensitive to depends dominantly on the number of expected background events rather than on the uncertainty of the background estimate.

The translation of the number of signal events into a dark shower production cross section is potentially subject to large systematic uncertainties, which are difficult to estimate from Monte Carlo simulations alone. Here we estimate the expected limit on the dark shower production cross section $\sigma_{\exp}^{95}$ using the nominal performance of the dark shower tagger, emphasizing that the tagger may show a different performance when applied to real data (see discussion below). The improvement in the expected limit $\sigma_{\exp}^{95}$ achieved by the dark shower tagger is shown in the right panel of figure 7 for the signal region EM4. Note that the cross section limit also takes the additional signal rejection caused by the tagger into account. The event numbers and the expected limit for the optimal tagging threshold (corresponding to $\epsilon_S = 0.13$) are compiled in table 3.

We can then translate $\sigma_{\exp}^{95}$ into a limit on the dark sector model coupling $g_q$, i.e. the coupling between the $Z'$ mediator and the SM quarks. We use that $\sigma(pp \to q_d \bar{q}_d) \propto g_q^2 \, \text{BR}(Z' \to q_d \bar{q}_d)$, which holds as long as the $Z'$ resonance is sufficiently narrow. For each $Z'$ mass we determine the limit based on the signal region that is most sensitive without a dark shower tagger. In the mass range considered here these are EM4 for $1 \text{ TeV} \lesssim m_{Z'} \lesssim 1.3 \text{ TeV}$ and EM2 for smaller $m_{Z'}$. Exploring larger $Z'$ masses would be computationally expensive, as it would require training the network on different signal regions. The expected limit is shown together with the existing LHC limits from ref. [10] in figure 8. We conclude that the use of a DGCNN classifier for semi-visible jets from dark showers has the potential to significantly improve the sensitivity of the search. For our benchmark model, in particular, the DGCNN tagger may allow to probe dark sector model couplings in a region of parameter space where conventional searches without neural network classifiers fail and where searches for displaced vertices are not yet sensitive.

We emphasise that the analysis presented above is based on the selection cuts of an existing mono-jet analysis, which is not optimised for dark showers. Hence, even greater improvements in sensitivity can be expected when combining the background reduction achieved by the tagger for semi-visible jets with relaxed cuts on the overall event topology. Of particular importance in the context of semi-visible jets is the cut on the separation angle $\Delta\phi$ in the azimuthal plane between the missing energy vector and any of the leading jets. In events where two semi-visible jets are produced back-to-back one typically obtains $\Delta\phi \approx 0$, unless one of the jets remains fully invisible. However, conventional mono-jet analyses require $\Delta\phi$ to be sufficiently large to suppress backgrounds from mismeasured jets. If such mismeasured jets can be reliably distinguished from semi-visible jets using deep neural networks, the cut on $\Delta\phi$ could be relaxed, which would significantly enhance the acceptance for semi-visible jets. Accurate simulations of this particular background are however very challenging, and we leave a study of the potential sensitivity of such a search to future work.

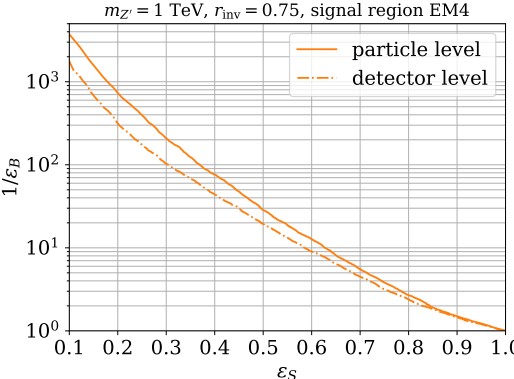
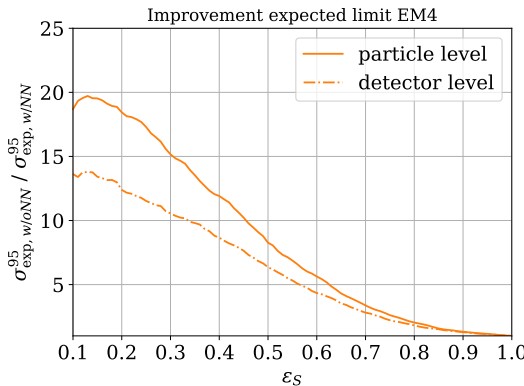

Figure 7: Left: Event-level ROC curves for a mono-jet search including a DGCNN dark shower tagger. Shown are the additional background event rejection and signal event efficiency relative to the search without a tagger. Right: Corresponding improvement of the expected limit on the dark shower production cross section. The jet constituents used as input for the DGCNN are either at the particle (solid lines) or detector level (dash-dotted lines).

## 5  Conclusions

Dark sectors with new strong dynamics may reveal themselves at the LHC in the form of dark showers resulting from the fragmentation and hadronisation of dark quarks. If some of the dark mesons in the shower are stable on cosmological scales (potentially explaining dark matter), while other dark mesons decay on collider scales, such dark showers lead to semi-visible jets. Although semi-visible jets depend on a number of dark sector parameters, such as the fraction of invisible particles and the mass of the dark mesons, in practice they often resemble QCD jets and are challenging to distinguish from backgrounds. Novel jet classification techniques are hence essential to enhance the sensitivity of the LHC to strongly interacting dark sectors.

In this paper we have explored the potential of supervised deep neural networks to identify semi-visible jets. As specific benchmark we have considered a scenario with GeV-scale dark mesons produced via a heavy vector mediator with mass in the TeV range; such a scenario is motivated by cosmological and astrophysical considerations and at the same time leads to a sizeable cross section for events with semi-visible jets at the LHC. We have compared three different types of neural network architectures: a convolutional neural network working on jet images, a Lorentz layer network based on an ordered set of four-momenta of jet constituents, and a dynamic graph convolutional neural network operating on particle clouds, i.e. an unordered set of jet constituents. While these three different neural network techniques deliver comparable results for the classification of top jets, we find that their performance differs notably in the more challenging classification of semi-visible jets. In particular, by dynamically updating the relation between jet constituents the graph neural network is able to learn more abstract features of a jet and outperforms the image-based convolutional and the Lorentz layer networks that we have considered.

We have then studied how the performance of the dynamic graph network changes as we vary the parameters of the strongly interacting dark sector and the corresponding semi-visible jets. As long as the same parameters are used for training and testing, the dark meson and mediator masses have no strong effect on the classification performance, while semi-visible jets with a larger fraction of invisible particles are in general easier to distinguish from QCD.

Table 3: Expected number of background events with systematic errors and expected limit on the number of signal events in the signal region EM4 at an integrated luminosity of 36.1 fb$^{-1}$. Listed are the event numbers for the mono-jet search without a dark shower tagger, with a DGCNN tagger operating on particles, and with a DGCNN tagger operating on detector level objects. For each case we also state the corresponding improvement of the expected limit on the dark shower production cross section. The DGCNN was trained and tested on a dark shower signal with our benchmark parameters (see main text).

|  | $B$ | $S_{\text{exp}}^{95}$ | $(\sigma_{\text{exp}}^{95})^{\text{w/oNN}}/\sigma_{\text{exp}}^{95}$ |
|---|---|---|---|
| without DGCNN tagger | $27640 \pm 610$ | 1239 | 1 |
| with DGCNN tagger (particle level) | $12.1 \pm 0.3$ | 8.2 | 19.7 |
| with DGCNN tagger (detector level) | $27.7 \pm 0.6$ | 11.7 | 13.8 |

However, the values of the dark sector parameters are not known a priori, and we find that the performance of the network significantly deteriorates when different parameters are used for training and testing. To mitigate this model dependence, we have trained the network on a mixed sample, which contains semi-visible jets with varying properties. This approach yields a more general classifier, which performs significantly better when applied to a range of model parameters.

Finally, we have shown how the sensitivity of the LHC to dark showers can be substantially enhanced by applying a jet classifier based on a dynamic graph neural network. For this purpose we have considered an existing ATLAS search for mono-jets, which is sensitive to events with one fully invisible dark shower and one semi-visible jet. We have then estimated the sensitivity that can be achieved by integrating our graph network into the analysis as a dark shower tagger. For our benchmark scenario we find an improvement in the sensitivity of more than an order of magnitude, leading to a significantly improved expected limit on the couplings of the model. The background reduction from tagging semi-visible jets may allow to relax cuts on the overall event topology and thereby further improve the sensitivity.

Various directions for future research on detecting dark showers with deep learning methods should be pursued. While we could not identify a particular observable that would control the classification performance of the neural network, more work is needed to explore what the network actually learns and how the choice of input features may further enhance the network performance. We refer to refs. [62–68] for examples of deep learning architectures that incorporate specific physics features to guide event classification.

In supervised learning, we rely on Monte Carlo events for the training, and it is crucial to avoid that the classification performance is biased by Monte Carlo artefacts. One should thus try to incorporate systematic uncertainties that account for the approximate modelling of a semi-visible jets, see e.g. [69–72], or further improve the Monte Carlo predictions for observables that drive the event classification of such subtle signatures. Likewise it will be important to asses the effect of pile-up removal on the performance of the neural network. We leave this question to future work.

Last but not least, one would like to use semi-supervised or unsupervised learning methods for the identification of dark shower events. For example, unsupervised machine learning algorithms based on autoencoders have successfully been used to search for anomalous jet substructure, see e.g. [73–76]. However, we find that it is not straightforward to apply this technique to the detection of semi-visible jets. Since the semi-visible jets often contain less

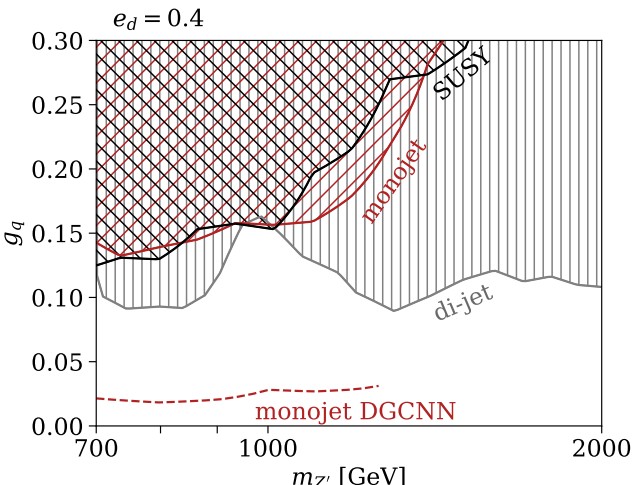

Figure 8: Expected limit on the benchmark model considered in this work from a mono-jet search [46] including a dynamic graph convolutional neural network dark shower tagger (labelled "monojet DGCNN"). The couplings of the $Z'$ mediator to dark quarks and SM quarks are denoted by $e_\mathrm{d}$ and $g_q$, respectively. Other LHC limits are taken from ref. [10].

information and structure than the QCD background jets, an autoencoder trained to reconstruct QCD may also be able to reconstruct semi-visible jets and thus may not detect semi-visible jets as an anomaly. Adapting the autoencoder approach for the detection of simple jet structures, and exploring alternative unsupervised and semi-supervised deep learning techniques [72, 77–80] for the identification of dark shower events, is left for future work.

# Acknowledgements

We thank Martin Erdmann, Silvia Manconi, Tilman Plehn, Jennifer Thompson and Patrick Tunney for discussions. This work is funded by the Deutsche Forschungsgemeinschaft (DFG) through the Collaborative Research Center TRR 257 "Particle Physics Phenomenology after the Higgs Discovery" and the Emmy Noether Grant No. KA 4662/1-1. Simulations were performed with computing resources granted by RWTH Aachen University under project thes0678.

# A   Jet and event generation

In this appendix, we first describe the generation of the signal and background jets used to train and test the networks in section 3. We then give details on the event generation for the search in section 4.

   To simulate the dark shower jets, we generate leading-order parton-level events for the dark quark production process $pp \rightarrow q_\mathrm{d}\bar{q}_\mathrm{d}$ at a collider energy of 14 TeV with MADGRAPH5_AMC@NLO [81] using the NN23LO1 PDF set [82] and a UFO file for the model introduced in section 2 implemented with FEYNRULES [83]. Renormalisation and factorisation scales are set to the default dynamic scale choice of MADGRAPH5_AMC@NLO. The samples are MLM-matched [84] with up to one additional hard jet, setting the matching scale to $x_\mathrm{qcut} = 100$ GeV. Shower and hadronisation are performed with PYTHIA 8 [85]. We employ PYTHIA's Hidden Valley module [14], which is adapted to the simulation of the dark shower

and of dark meson production as detailed in [10]. The running of the dark coupling $\alpha_d$ is determined by the confinement scale $\Lambda_d$. If not explicitly stated otherwise we use the default parameters $m_{Z'} = 1$ TeV, $m_\pi = m_\rho = \Lambda_d = 5$ GeV. The couplings of the $Z'$ mediator to dark quarks and SM quarks are set to $e_d = 0.4$ and $g_q = 0.1$, respectively. These couplings enter the $Z'$ width, but their value is not relevant for the production of the training and test sets of the dark shower jets.

The light QCD background jets are obtained from di-jet events generated at leading order at a collider energy of 14 TeV with MADGRAPH5_AMC@NLO using the NN23LO1 PDF set. Renormalisation and factorisation scales are set to the default dynamic scale choice of MAD-GRAPH5_AMC@NLO. Shower and hadronisation are performed with PYTHIA 8.

In both the dark shower signal and the background event samples, we employ the Fast-Jet [86] implementation of DELPHES 3 [60] to cluster fat jets using the anti-$k_T$ algorithm [87] with jet radius $R = 0.8$. No detector simulation is performed unless explicitly stated. When a detector simulation is included, it is performed with DELPHES 3 using the ATLAS detector card. To select jets originating from dark showers for the signal sample, we additionally require $\Delta R < 0.8$ for the angular distance $\Delta R = \sqrt{\Delta \eta^2 + \Delta \phi^2}$ between the jet axis and a truth-level dark quark. Otherwise jets from QCD initial state radiation would enter the signal samples of semi-visible jets. For the network comparisons in section 3 we use fat jets within the transverse momentum range $p_{T,\text{jet}} = 150 \dots 350$ GeV. To simulate samples with larger transverse momenta would be computationally more expensive since there is no generator cut which can be used to significantly enhance event generation in the high-$p_T$ tail.

For the classification of top-quark jets we use the benchmark dataset from ref. [37] to be able to compare the network performance for top-tagging with the results quoted in [37]. The dataset consists of jets from hadronically decaying tops and light QCD di-jets at a collision energy of 14 TeV, simulated with PYTHIA 8. Jets in the $p_T$ interval [550 GeV, 650 GeV] are clustered according to the anti-$k_T$ jet algorithm with a jet radius of 0.8, after a fast detector simulation with DELPHES 3. Jets are required to fulfil $|\eta|_{\text{jet}} < 2$. For the top jets, a parton level top is required to fall within $\Delta R = 0.8$ of the final jet. Additionally, the three quarks from the hadronic top decay at tree-level are required to obey $\Delta R < 0.8$ with respect to the top.

For all jet samples, the four-momenta of the 200 constituents with highest $p_T$ are stored in descending $p_T$ order. For jets with fewer constituents, zeros are added to obtain the same array size for each jet.

For the search discussed in section 4, the samples for training the jet tagger are produced with the tool chain described above. Signal events are generated at a collision energy of 13 TeV for the dark sector benchmark model with $r_{\text{inv}} = 0.75$, $m_{\pi_d} = 4$ GeV and $m_{\rho_d} = \Lambda_d = 5$ GeV. The QCD jets are extracted from $Z$+jets events produced at 13 TeV including MLM matching with up to two hard jets. We generate two different samples which populate the fiducial volume of the signal regions EM2 and EM4 defined in ref. [46], respectively. The samples consist of 200k signal and 200k background jets.

To test exclusion for different parameter points, we generate events as described above to determine the cut and tagging efficiencies. Note that much fewer events have to be generated compared to the large event number needed to extract the jet sample for training.

# B  Neural network architectures

In this appendix we present the architectures for the convolutional neural network (CNN), the Lorentz-layer neural network (LoLa), and the dynamic graph convolutional neural network (DGCNN) used for the classification of semi-visible and top jets in the main part of the paper.

We use KERAS 2.3.1 [88] with TENSORFLOW 1.13.1 [89] as backend for the implemen-

tation, training and evaluation of our networks. If not stated differently, we use the ADAM optimiser [90] in its default configuration to optimise the categorical cross entropy loss. The categorical cross entropy for one-hot encoded labels is given by

$$CE(y_{\text{true}}, y_{\text{pred}}) = -\sum_{i=1}^{2} y_{\text{true},i} \ln(y_{\text{pred},i}) = -\ln(y_{\text{pred,true}}).$$

Here, $y_{\text{true}}$ and $y_{\text{pred}}$ correspond to the true labels and the predicted labels, respectively. Since labels are one-hot encoded, $y_{\text{true},i}$ is equal to zero for the wrong class and equal to one for the correct class.

## Convolutional neural networks

We use a CNN consisting of several convolutions for feature extraction and maximum pooling layers for dimensionality reduction, followed by a fully connected part for classification. The convolutional layers use 128 filters with kernels covering 3x3 pixels, the pooling layers apply the max-function on 2x2 pixel windows with stride two, reducing the dimension of the image along both axes by a factor of two. Using the max function for pooling implicitly assumes that pixels with higher intensity are more important for the classification.

The activation function of choice throughout the network is ReLu, only the last layer applies softmax activation. The output consists of two nodes, one for each class. A sketch of the architecture is shown in the left panel of figure 9. We have confirmed that varying the network architecture does not improve the performance for semi-visible jets.

To obtain a jet image from the four-momenta of its constituents, we first calculate the pseudo-rapidity $\eta$, azimuthal angle $\phi$ and the transverse momentum $p_T$ of each constituent. The following preprocessing steps are applied [36]: (1) The $p_T$-weighted centroid of the jet is shifted to the origin in the $\eta$-$\phi$-plane. (2) All constituents are rotated such that the $p_T$-weighted principal axis points in the $\eta$-direction. (3) The axes is flipped such that the maximum intensity (sum of $p_T$) is in the upper right quadrant. (4) The jet image is generated as the $p_T$ weighted histogram in $\eta$ and $\phi$ and normalised by dividing by the total $p_T$. We use 40 bins within the interval [-0.8, 0.8] for both $\eta$ and $\phi$.

For the results shown in this appendix, we train the network on the training set for top tagging provided in [37] to be able to compare with the results presented in [37]. The set consists of 600k jets for each class. The maximum number of epochs is set to 100. The learning rate is reduced, if the validation loss does not improve for three epochs, and the training is stopped after five more epochs without improvement. The network performance results in this section are based on the independent test of 200k background and 200k signal jets provided along the benchmark data set.

## Lorentz-layer neural networks

Following ref. [37] we construct a network based on a so-called combination layer (CoLa) followed by a Lorentz layer (LoLa). The CoLa receives a list of particle four-momenta, ordered in $p_T$, and calculates a number of linear combinations of those vectors. The coefficients in these linear combinations are trainable, and the output of this layer consists of the original list of momenta appended by the learned linear combinations. The LoLa then transforms every four-vector into

$$\tilde{k}_j \rightarrow k_j = \begin{pmatrix} m^2(\tilde{k}_j) \\ p_T(\tilde{k}_j) \\ w_{jm}^{(E)} E(\tilde{k}_m) \\ w_{jm}^{(d)} d_{jm}^2 \end{pmatrix}. \tag{3}$$

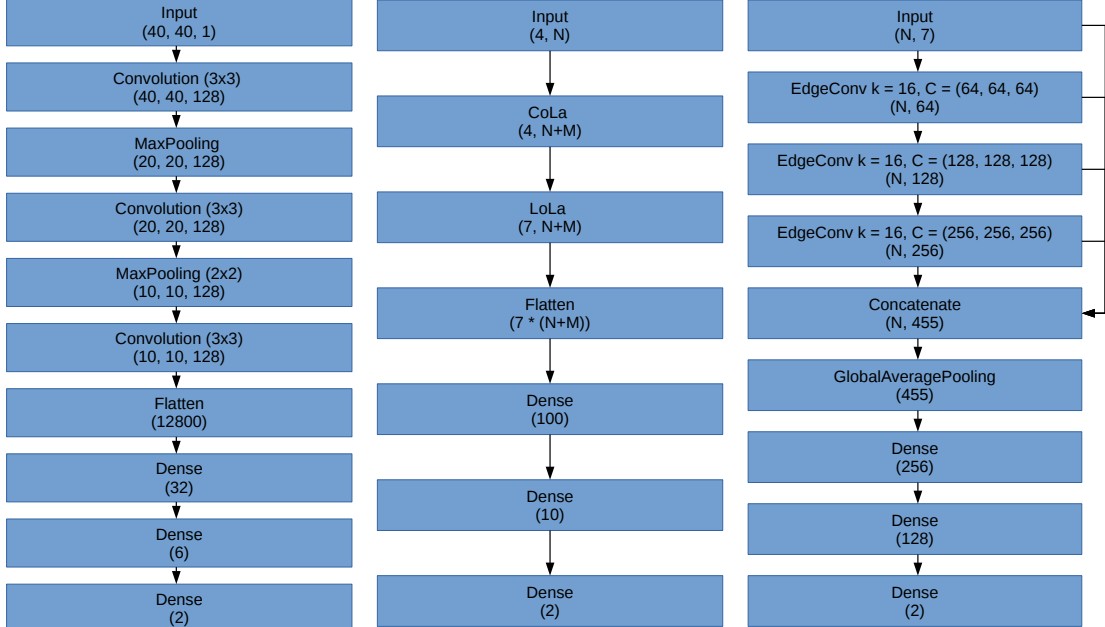

Figure 9: Sketch of the CNN architecture for jet image classification (left panel), the LoLa architecture for classification on four-vectors (middle panel) and for the DGCNN architecture for classification on particle clouds (right panel). Each block corresponds to one layer in the network. The first line of each block describes the kind of layer and the kernel size, if applicable. The first line in the blocks for EdgeConv layers give the number of nearest neighbours k and the number of filters used in the three convolutions C. The second line states the output dimension of each layer, with N the number of jet constituents used, and M the number of added linear combinations in the LoLa network.

The first entry corresponds to the invariant mass and the second entry to the transverse momentum of the particle. The third entry is a linear combination of the energies of all particles weighted by trainable parameters. The last entry is a trainable combination of Minkowski distances between particle four-momenta. In practice four distance combinations are added to the vector. For two of the added entries, we sum over the index m, while we take the minimum for the other two entries. To obtain a classification, the output of the LoLa is flattened and passed on to a fully connected network. We use ReLu as activation for the fully connected layers, except for the classification output, where we apply softmax. CoLa and LoLa do not apply activation functions. A sketch of the architecture is displayed in the central panel of figure 9. We have confirmed that varying the network architecture does not improve the performance for semi-visible jets.

Training is performed in the same way as for the CNN, including the learning rate schedule and early stopping.

The performance of the CoLa/LoLa architecture depends on the number of jet constituents that is used as input for the network and on the number of linear combinations added by the CoLa. Ordering the particles in descending $p_T$, we find that the best performance is achieved with about 40 jet constituents. The network performance is not particularly sensitive to the number of linear combinations in the CoLa. We have chosen 40 constituents and 15 linear combinations, consistent also with ref. [37].

**Dynamic graph neural networks**

Dynamic graph convolutional neural networks (DGCNNs) have been introduced in ref. [38] and applied to jet tagging in ref. [39]. These network architectures operate on point clouds with so-called edge convolution (EdgeConv) layers. For jet-tagging the point cloud consists of particles, i.e. the jet constituents.

The edge convolution differs from a convolution over an image in the definition of the local patch that the kernel observes. In an image, the local patch corresponds to some neighbourhood of pixels. For an edge convolution, a local graph is constructed for each point in the cloud by finding its k nearest neighbours with respect to some metric which has to be specified. The corresponding graph is called a k-nearest-neighbour (knn) graph. For each particle the convolution is then performed over its nearest neighbours, i.e. $x_i' = \Omega_{j=1}^k h_\Theta(x_i, x_j)$. Here, $x_i$ corresponds to the $i$-th point in the cloud and $x_i'$ to the output of the convolution at this point. The kernel $h_\Theta(x_i, x_j)$ is implemented as a fully connected layer and calculates edge features for a point and each of its k neighbours. Those k edge features are reduced to one output feature vector $x'$ by the aggregation function $\Omega$. This function should not depend on the order of inputs. We use the mean in this work. The same $h$ is then used on all points and their neighbours, just like the kernel in a regular convolution. We follow ref. [39] and use $h_\Theta(x_i, \Delta x_j)$, where $\Delta x_j$ is the difference of the features of $x_i$ and $x_j$.

Since the EdgeConv operation produces as output again a point cloud with the same number of points as the input, one can stack EdgeConvs onto each other. Note that the number of output features for the particles is variable and changes from layer to layer. Calculating new nearest neighbours from the output of the previous EdgeConv allows for points that are initially far apart to be grouped close in feature space.

We follow ref. [39] in selecting the following input features for the DGCNN:

1. $\Delta\eta = \eta - \eta_{\text{jet}}$ where $\eta$ ($\eta_{\text{jet}}$) is the rapidity of the constituent (the jet),

2. $\Delta\phi = \phi - \phi_{\text{jet}}$ where $\phi$ ($\phi_{\text{jet}}$) is the azimuthal angle of the constituent (the jet),

3. $\log(p_T)$ - constituent's transverse momentum in GeV,

4. $\log(p_T/p_{T_{\text{jet}}})$ - constituent's $p_T$ relative to the jet $p_T$,

5. $\log(E)$ - constituent's energy in GeV,

6. $\log(E/E_{\text{jet}})$ - constituent's energy relative to the jet energy,

7. $\Delta R = \sqrt{\Delta\eta^2 + \Delta\phi^2}$.

We use a combination of EdgeConv layers followed by fully connected layers for the classification of particle clouds. First, we construct three EdgeConv blocks. At the beginning of each block a new k-nearest-neighbours graph is generated. We set $k = 16$ for all blocks. In the first block, the distance between particles is calculated only in $\eta$ and $\phi$. In the later blocks, the distance is calculated as the euclidean distance of the complete feature vector. Each EdgeConv block consists of three convolutions on the constructed graph with the same number of filters. The number of convolution filters corresponds to the number of features for each particle in the next layer. We use 64 filters in the first block, 128 in the second and 256 in the third. The increasing number of filters allows the network to extract more and more detailed features.

We concatenate the input features and the features from each EdgeConv block for each particle, so that we end up with $7 + 64 + 128 + 256 = 455$ features for each jet constituent after the EdgeConv layers of the network. Since we want to keep the network independent of the ordering of the particles, we need to aggregate the constituents in a way that is invariant under permutation. We choose to use the average feature vector, since it shows better results

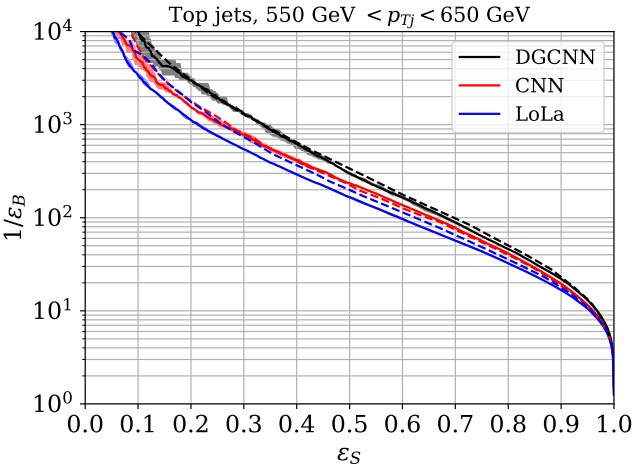

Figure 10: ROC curves for top tagging with the different neural networks architectures described in this appendix. The dashed lines denote the ROC curves of the corresponding architectures as presented in ref. [35].

than, for example, the max aggregation. This results in one feature vector with 455 features. These 455 features are the input to the fully connected part of the network. We use three fully connected layers with 256, 128 and 2 nodes respectively and adopt ReLu activation for each layer except for the classification output, where we apply softmax.

To prevent the fully connected part of the network from overfitting, we use dropout layers in front of the first two fully connected layers for regularisation and update only 90% of the weights. A sketch of the architecture is shown in the right panel of figure 9.

As before, we use the training and test set for top tagging provided in ref. [35]. We use a learning rate schedule during training. The initial learning rate is set to $3 \times 10^{-4}$. We increase it linearly for 8 epochs to $3 \times 10^{-3}$ and decrease it to its initial value within another 8 epochs. The next 4 epochs we reduce the learning rate further to $5 \times 10^{-7}$. Such a learning rate schedule is supposed to lead to faster convergence [91]. Training finishes after 20 epochs. We do not perform a dedicated hyperparameter optimisation in this work. The parameters we use for this network are comparable to those in ref. [39], except for the number of jet constituents which we fix to 40 for comparability with the LoLa network. We note that different hyperparameter settings could be optimal for top tagging or for tagging semi-visible dark jets. This optimisation is left for future work.

**Network performances**

Figure 10 compares the ROC curves (see section 3.1) for top tagging with the CNN, the LoLa and the DGCNN, respectively, to the results presented in ref. [35]. As in section 3.1 the stability and reproducibility of the network performance is evaluated by training five networks with independent random weight initialisation. The small spread in performance indicated by the shaded band around the ROC curves shows that the training convergence of the network is stable.

We find very good agreement between our results and the results presented in ref. [35] for the CNN and DGCNN, and a reasonable agreement for the LoLa network. Since the DGCNN provides the best performance, we focus on this architecture in this work and do not attempt to further optimise the performance of the LoLa network.

We compare the number of parameters, the inference time and the required storage for

Table 4: Comparison of the number of parameters, the inference time and the storage needed for the different architectures introduced in this section.

| Network | Parameters | Inference time [$\mu s$] | Needed storage [MB] |
|---|---|---|---|
| CNN | 706,292 | 19.1 | 8.2 |
| CoLa/LoLa | 48,031 | 3.59 | 0.61 |
| DGCNN | 411,458 | 141.1 | 4.8 |

the different networks in table 4. The inference time of the networks may, for example, be crucial when using them as event triggers and also affects the computational effort needed for training. To obtain the inference time, we predict the output of 400k jets with a batch size of 512. We test the networks on a NVIDIA Tesla V100-SXM2-16GB GPU and display the average time needed per image. While the CoLa/LoLa consists mostly of hand crafted, hard coded features and thus involves comparably few trainable parameters, the large number of filters in the convolutions results in a much larger number of trainable parameters for the CNN. Also the DGCNN has many filters in the convolutions performed in the EdgeConv blocks, and thus significantly more parameters than the CoLa/LoLa network. The inference time is largest for the DGCNN, even though it has fewer parameters than the CNN. This is due to the number of calculations needed specifically to compute the pairwise distance for all particles and to construct the k-nearest-neighbours graph.

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
