# Peer review of "Casting a graph net to catch dark showers"

_SciPost Physics, doi:SciPost Phys. 10, 046 (2021)_

## Round 1 · Referee Report · Anonymous (Referee 1) · 2020-7-21

Report
This paper demonstrates the application of graph networks applied to the jets of dark showers at the LHC. The work is timely, the analysis is thorough, and the paper is well-written. I congratulate the authors on this nice addition to the literature! Below are a small number of minor comments that it would be good to address before the paper is published.
- [36,48,49] block on p5, might be best to add 1902.09914.
- p5: "dense neural network" is Tensorflow jargon - perhaps "fully connected network" would be more appropriate.
- Discussion on p6, before Sec. 3.1: I'm not sure how this is different than a CNN; a CNN operator, just like the EdgeConv operator is local, but having many layers allows for global relations. Please either tell me why this is wrong or modify the text accordingly.
- "Normally, the network prediction is given by the class with the highest probability." -> I would never advocate to do this, since the best cut should depend on the relative abundance of signal and background which most certainly is not 50%-50% (=the prior used in your training).
- Fig. 3 right: can you please comment how this compares with the results shown in the community top tagging paper (1902.09914)?
- Why R = 0.8? That is not the radius used by ATLAS.
- p12: Please explain 3.84 (I am sure you have a good reason, but please don't make the reader guess!).
- "We can therefore safely neglect additional systematic uncertainties introduced by the dark shower tagger." -> the stat. uncertainty is at most 30% - it does not seem crazy to me that the systematic uncertainty would be comparable. Why is it then save to neglect?
- Last paragraph of the conclusions: unsupervised methods are not the only anomaly detection procedures that have been proposed - there are a variety of semi-supervised methods that may be (more) effective. See e.g. https://iml-wg.github.io/HEPML-LivingReview/.
- Delphes: what detector setup are you using?

---

## Round 1 · Referee Report · Anonymous (Referee 2) · 2020-8-4

Strengths
1) Comparison of different network architectures 2) Exploration of model parameters affecting network performance 3) Forecast of limits
Weaknesses
1) Explanation of model
Report
This is a very strong paper and I suggest it for publication. The authors examine different supervised neural network architectures to search for semi-invisible jets. They find that a Dynamic Graph Convolutional Neural Network performs better than jet images or Lorentz Layer networks.
I have a few minor comments and requested changes.
1) In section 2, the authors describe the model and how it can lead to semi-invisible jets. While the diagram is nice, showing how the Z' decays, it would be very helpful to explicitly show the decay modes of the ${\rho^+}$, $\rho^0$. This would help explain why the $\rho^0$ is the only dark meson decaying on collider time scales and setting the R_inv = 0.75. For instance in [1809.10184], the $\rho^+$ decays promptly to q q$^{\prime}$ when $m_{\rho} < 2 m_{\pi}$.
2) In Figure 3 the CNN and LoLa change which is better for tops versus semi-visible. Do the authors have any insights into this?
3) Table 2 shows the results for training and testing on different meson masses. However, they never show results for testing on a lighter mass than was trained on, is there a reason for this.
4) Using a parameterized network would greatly help over the mixed samples. These allow the network to generalize between samples better. The authors mention this but do not cite the paper showing this [https://arxiv.org/abs/1601.07913 for hep]
5) On page 12, the number 3.84 seems to come out of nowhere. Rephrasing such that they are looking for the 95% CL exclusion, which is when q_mu = 3.84 would help the reader.
6) On page 13, there needs to be more explanation about why systematics can now safely be neglected, especially since the previous section was about how training on the wrong parameters (even on mixed samples) leads to a reduction in performance.
7) They should also cite the few other hep-ph papers utilizing graph neural networks such as https://arxiv.org/abs/1807.09088, (https://iml-wg.github.io/HEPML-LivingReview)

---

## Round 1 · Referee Report · Anonymous (Referee 3) · 2020-8-5

Report
Requested changes
1- The high level variables referred to at the bottom of p. 2 are conventionally referred to collectively as $N$-subjettiness, not subjettiness. 2- Given that baryons would be expected to have a mass $\sim \Lambda_d$, the same as the assumption made for the mesons in the text, better justification for ignoring baryons in the phenomenological study should be given. Since an $SU(3)$ gauge group is assumed, I would expect a combinatoric suppression of $\sim 1/9$ in collider production, as in the SM. The authors should comment on whether this is enough to ignore the baryons for their purposes or if additional suppression must be provided. 3- Since no hyperparameter optimization is performed for the networks going from top to semi-visible jet tagging, can the authors exclude the possibility that the performance differences in the networks would be reduced for a difference choice of hyperparameters? 4- While the DGCNN showed the best performance for a given choice of parameters when trained on those parameters, when ultimately trained with mixed samples (a much more realistic scenario) the performance is degraded by a factor of a few. Have the authors checked that the other architectures degrade similarly when training on mixed samples? If so, this should be stated; if not, it should be done. Perhaps the other architectures are more robust to model variation. 5- I want to second the other referee's concern about ignoring systematic uncertainties induced by the tagger. The uncertainty in the signal efficiency in real world events for a tagger trained on simulated signal where no calibration region exists can be significant and should be discussed. 6- Some measure of the effort of training the various networks in Table 4, perhaps wall clock time on equivalent systems, would be helpful as a metric in benchmarking the ultimate network performance in the absence of hyperparameter optimization.

---

## Round 1 · Referee Report · Anonymous (Referee 4) · 2020-8-7

Strengths
- This work is an application of DL techniques to the work [10], which is interesting.
Weaknesses
- This work is an application of "known" DL techniques on a specific physics example.
Report
Requested changes
-
Learnings with low level information (particle level) generally suffer from pileups, especially dealing with jets. There are prescriptions to reduce effects from pileups. How much is DGCNN robust under these conventional procedures for pileups removals ?
-
It would be appreciated if authors perform comparison analyses between semi-invisible and hadronic tau. On top of this, W+jets (where W->hadronic tau+neutrino) study would be relevant for mono-jet study.
-
Authors "observe" performance of various DL techniques on a subject of "semi-invisible jet" (just like HEP-EX.) It would be very welcomed if authors provide some reasons (qualitative analyses) behind these performance dependency on Dark QCD model-parameters.
-
I guess that the Z' mass dependence in analysis is negligible as authors seem like to fix PT range of jet.
- For Z' = 2TeV where PT of jet are O(1) TeV, what would be expected performance over top-jet with O(1) TeV PT ?

---

## Round 2 · Referee Report · Anonymous · 2020-11-8

Report
Thank you for the new version and for taking into account my comments and suggestions. I think the paper is nearly ready for publication. Below are some ultra minor suggestions.
Comment 8: "We can therefore safely neglect additional systematic uncertainties introduced by the dark shower tagger." -> the stat. uncertainty is at most 30% - it does not seem crazy to me that the systematic uncertainty would be comparable. Why is it then save to neglect?
Reply 8: We are grateful for the reviewer to raise this important point. We assume that it will still be possible to determine the dominant backgrounds with data-driven methods, like using Z(->mu mu)+jets to estimate Z(->nu nu)+jets. The systematic uncertainties in the performance of the dark shower tagger should then not affect the background estimates, only the signal efficiency. We have substantially rewritten the text to make this point clear.
Comment on the reply: I agree that one can use Z->mumu to constrain Z->nunu, but there will still be some non-negligible systematic uncertainty. For example, the Z->mumu sample will have significantly lower stats than the neutrino sample and this stat. uncertainty will become a systematic uncertainty on the Z->nunu prediction. I don't disagree with what is written, but just because something can be constrained with data doesn't mean the uncertainty will be negligible. It might be worth adding a small comment about this.
Comment 9: Last paragraph of the conclusions: unsupervised methods are not the only anomaly detection procedures that have been proposed - there are a variety of semi-supervised methods that may be (more) effective. See e.g. https://iml-wg.github.io/HEPML-LivingReview/.
Reply 9: We now also mention semi-supervised methods.
Comment on the reply: I see you added the word "semisupervised" which is great, but you may also want to consider adding citations, as these approaches will likely have complementary challenges to the ones you mention for the unsupervised methods (and note that [72] is a mix of semi-supervised and unsupervised training so it definitely fits in both sets of citation blocks!).

---

## Round 2 · Referee Report · Anonymous · 2020-12-24

Report
I am happy with the authors' changes in response to my comments and those of the other referees, and am glad to recommend moving toward publication, once the minor error given bellow is corrected.
I do however have a minor comment on the new last paragraph in the conclusions, which the authors may wish to take under consideration. I think the authors make a good case for the difficulty of applying unsupervised methods to the extraction of BSM physics in hadronic final states where the jets are "simpler" than QCD by some measure. I do not think their point applies to more semi-supervised methods, such as those of [arXiv:1805.02664, 2001.04990]. In fact, the types of $Z'$ searches discussed in the text are a fairly natural fit for the machine-learning-assisted generalized sideband procedures discussed by the papers above and related work.
Requested changes
1- Refs. [36] and [87] are identical. Please merge.

---

## Round 2 · Author Response

List of changes
Report 3:
Comment 1: [36,48,49] block on p5, might be best to add 1902.09914.
Reply 1: We have added the reference (which was already cited elsewhere in the paper).
Comment 2: p5: "dense neural network" is Tensorflow jargon - perhaps "fully connected network" would be more appropriate.
Reply 2: We have changed this everywhere.
Comment 3: Discussion on p6, before Sec. 3.1: I'm not sure how this is different than a CNN; a CNN operator, just like the EdgeConv operator is local, but having many layers allows for global relations. Please either tell me why this is wrong or modify the text accordingly.
Reply 3: We have modified the text to clarify the difference between a CNN and a dynamical graph convolutional neural network.
Comment 4: "Normally, the network prediction is given by the class with the highest probability." -> I would never advocate to do this, since the best cut should depend on the relative abundance of signal and background which most certainly is not 50%-50% (=the prior used in your training).
Reply 4: We have removed this sentence and rewritten the following one.
Comment 5: Fig. 3 right: can you please comment how this compares with the results shown in the community top tagging paper (1902.09914)?
Reply 5: This comparison is provided in the appendix (figure 10). We have added a corresponding sentence.
Comment 6: Why R = 0.8? That is not the radius used by ATLAS.
Reply 6: We use different jet definitions for the event selection and for the dark shower tagging. We have modified the text to clarify this point.
Comment 7: p12: Please explain 3.84 (I am sure you have a good reason, but please don't make the reader guess!).
Reply 7: We have added an explanation.
Comment 8: "We can therefore safely neglect additional systematic uncertainties introduced by the dark shower tagger." -> the stat. uncertainty is at most 30% - it does not seem crazy to me that the systematic uncertainty would be comparable. Why is it then save to neglect?
Reply 8: We are grateful for the reviewer to raise this important point. We assume that it will still be possible to determine the dominant backgrounds with data-driven methods, like using Z(->mu mu)+jets to estimate Z(->nu nu)+jets. The systematic uncertainties in the performance of the dark shower tagger should then not affect the background estimates, only the signal efficiency. We have substantially rewritten the text to make this point clear.
Comment 9: Last paragraph of the conclusions: unsupervised methods are not the only anomaly detection procedures that have been proposed - there are a variety of semi-supervised methods that may be (more) effective. See e.g. https://iml-wg.github.io/HEPML-LivingReview/.
Reply 9: We now also mention semi-supervised methods.
Comment 10: Delphes: what detector setup are you using?
Reply 10: We have added this information (when we perform a detector simulation, we use the ATLAS card, in particular for the mono-jet analysis).
Report 4:
Comment 1: In section 2, the authors describe the model and how it can lead to semi-invisible jets. While the diagram is nice, showing how the Z' decays, it would be very helpful to explicitly show the decay modes of the ρ+,ρ0. This would help explain why the ρ0 is the only dark meson decaying on collider time scales and setting the R_inv = 0.75. For instance in [1809.10184], the ρ+ decays promptly to q q' when mρ < 2 mπ.
Reply 1: Indeed, r_inv is model-dependent and it is important to consider the effect of varying it. For our benchmark choice r_inv = 0.75, we rely on the specific model from arXiv:1907.04346, where all SM quarks have the same U(1)' charge and hence the charged dark rho mesons cannot decay into a quark-antiquark pair. We feel that the present paper is not the right place for discussing these details and have therefore decided to give only a brief summary and refer the reader to earlier publications.
Comment 2: In Figure 3 the CNN and LoLa change which is better for tops versus semi-visible. Do the authors have any insights into this?
Reply 2: We have not investigated this effect further.
Comment 3: Table 2 shows the results for training and testing on different meson masses. However, they never show results for testing on a lighter mass than was trained on, is there a reason for this.
Reply 3: The effects when testing on lighter masses than the one used for training is completely analogous to when testing on heavier masses.
Comment 4: Using a parameterized network would greatly help over the mixed samples. These allow the network to generalize between samples better. The authors mention this but do not cite the paper showing this [https://arxiv.org/abs/1601.07913 for hep]
Reply 4: We have added this reference, which was indeed missing in our discussion.
Comment 5: On page 12, the number 3.84 seems to come out of nowhere. Rephrasing such that they are looking for the 95% CL exclusion, which is when q_mu = 3.84 would help the reader.
Reply 5: See above (report 3, reply 7).
Comment 6: On page 13, there needs to be more explanation about why systematics can now safely be neglected, especially since the previous section was about how training on the wrong parameters (even on mixed samples) leads to a reduction in performance.
Reply 6: See above (report 3, reply 8).
Comment 7: They should also cite the few other hep-ph papers utilizing graph neural networks such as https://arxiv.org/abs/1807.09088, (https://iml-wg.github.io/HEPML-LivingReview)
Reply 7: We have added a number of relevant reference.
Report 1:
Comment 1: The high level variables referred to at the bottom of p. 2 are conventionally referred to collectively as N-subjettiness, not subjettiness.
Reply 1: Changed as suggested.
Comment 2: Given that baryons would be expected to have a mass ∼Λd, the same as the assumption made for the mesons in the text, better justification for ignoring baryons in the phenomenological study should be given. Since an SU(3) gauge group is assumed, I would expect a combinatoric suppression of ∼1/9 in collider production, as in the SM. The authors should comment on whether this is enough to ignore the baryons for their purposes or if additional suppression must be provided.
Reply 2: If baryons are produced in the dark shower their only effect would be to slightly increase the fraction of invisible particles r_inv. Since we consider different values of r_inv, our results therefore do not rely on any specific assumption regarding the production of baryons in the dark shower. Nevertheless, the reviewer is correct that our benchmark choice r_inv = 0.75 assumes negligible contribution from baryons (we have now made this assumption explicit in the text). The reason for this assumption is that in the specific case we consider, the masses of dark mesons and dark baryons are determined not only by Λd, but also by the masses of the dark quarks, and hence it is to be expected that the baryons are somewhat heavier than the mesons. Nevertheless, we feel that the present paper is not the right place to discuss these details and have therefore decided to keep the discussion short. We also note that it is presently not possible to simulate the production of dark baryons with the Hidden Valley module of Pythia.
Comment 3: Since no hyperparameter optimization is performed for the networks going from top to semi-visible jet tagging, can the authors exclude the possibility that the performance differences in the networks would be reduced for a difference choice of hyperparameters?
Reply 3: We thank the reviewer for raising this point. We have tried a number of different ways of modifying the structure and hyperparameters of the CNN and the LoLa network, finding no improvement over the performance shown in our plots. We have added a corresponding comment.
Comment 4: While the DGCNN showed the best performance for a given choice of parameters when trained on those parameters, when ultimately trained with mixed samples (a much more realistic scenario) the performance is degraded by a factor of a few. Have the authors checked that the other architectures degrade similarly when training on mixed samples? If so, this should be stated; if not, it should be done. Perhaps the other architectures are more robust to model variation.
Reply 4: The performance of all three neural networks is substantially degraded when training on the wrong meson mass and slightly degraded when training on mixed samples. While the effect is largest for the DGCNN (reflecting the fact that this architecture is able to learn more details of the semi-visible jets), it still easily outperforms the other two network types for all combinations that we have considered. We have added footnote 3 to clarify this point.
Comment 5: I want to second the other referee's concern about ignoring systematic uncertainties induced by the tagger. The uncertainty in the signal efficiency in real world events for a tagger trained on simulated signal where no calibration region exists can be significant and should be discussed.
Reply 5: See above (report 3, reply 8).
Report 5:
Comment 1: Learnings with low level information (particle level) generally suffer from pileups, especially dealing with jets. There are prescriptions to reduce effects from pileups. How much is DGCNN robust under these conventional procedures for pileups removals ?
Reply 1: We thank the reviewer for this interesting question. We have started looking into the effects of pile-up, but the effort required for a conclusive study are beyond the scope of the present work. Nevertheless, we now mention this important direction for future work in the conclusions.
Comment 2: It would be appreciated if authors perform comparison analyses between semi-invisible and hadronic tau. On top of this, W+jets (where W->hadronic tau+neutrino) study would be relevant for mono-jet study.
Reply 2: For the lightest dark meson masses that we consider (m_meson = 5 GeV) and assuming that hadronic taus have not been included in the background sample, the network indeed does not reject hadronic taus as efficiently as QCD jets. However, hadronic taus are in principle easy to distinguish from semi-visible jets, as they have much fewer constituents. Since a simple cut on the number of constituents already removes most hadronic taus, we expect that training the network on a background sample containing hadronic taus will make it possible to efficiently suppress this background.
Comment 3: Authors "observe" performance of various DL techniques on a subject of "semi-invisible jet" (just like HEP-EX.) It would be very welcomed if authors provide some reasons (qualitative analyses) behind these performance dependency on Dark QCD model-parameters.
Reply 3: The discussion of model dependence in section 3.2 is intended precisely for this purpose, i.e. to identify the dependence of the performance of the network on the model parameters. For a more detailed discussion of how the properties of dark showers depend on the underlying model, we refer to the recent paper arXiv:2007.11597.
Comment 4: I guess that the Z' mass dependence in analysis is negligible as authors seem like to fix PT range of jet. For Z' = 2TeV where PT of jet are O(1) TeV, what would be expected performance over top-jet with O(1) TeV PT ?
Reply 4: The reason that we focus on relatively low pT is that most of the dark showers remain invisible. So the transverse momentum distribution of semi-visible jets falls steeply above a few hundred GeV even if the dark quarks are produced with TeV-scale energy. Generating samples of semi-visible jets with very high pT is hence computationally very expensive and also not very promising, given that the signal efficiency will be very low irrespective of the performance of the tagger.

---

## Round 2 · List of Changes

Report 3:
Comment 1: [36,48,49] block on p5, might be best to add 1902.09914.
Reply 1: We have added the reference (which was already cited elsewhere in the paper).
Comment 2: p5: "dense neural network" is Tensorflow jargon - perhaps "fully connected network" would be more appropriate.
Reply 2: We have changed this everywhere.
Comment 3: Discussion on p6, before Sec. 3.1: I'm not sure how this is different than a CNN; a CNN operator, just like the EdgeConv operator is local, but having many layers allows for global relations. Please either tell me why this is wrong or modify the text accordingly.
Reply 3: We have modified the text to clarify the difference between a CNN and a dynamical graph convolutional neural network.
Comment 4: "Normally, the network prediction is given by the class with the highest probability." -> I would never advocate to do this, since the best cut should depend on the relative abundance of signal and background which most certainly is not 50%-50% (=the prior used in your training).
Reply 4: We have removed this sentence and rewritten the following one.
Comment 5: Fig. 3 right: can you please comment how this compares with the results shown in the community top tagging paper (1902.09914)?
Reply 5: This comparison is provided in the appendix (figure 10). We have added a corresponding sentence.
Comment 6: Why R = 0.8? That is not the radius used by ATLAS.
Reply 6: We use different jet definitions for the event selection and for the dark shower tagging. We have modified the text to clarify this point.
Comment 7: p12: Please explain 3.84 (I am sure you have a good reason, but please don't make the reader guess!).
Reply 7: We have added an explanation.
Comment 8: "We can therefore safely neglect additional systematic uncertainties introduced by the dark shower tagger." -> the stat. uncertainty is at most 30% - it does not seem crazy to me that the systematic uncertainty would be comparable. Why is it then save to neglect?
Reply 8: We are grateful for the reviewer to raise this important point. We assume that it will still be possible to determine the dominant backgrounds with data-driven methods, like using Z(->mu mu)+jets to estimate Z(->nu nu)+jets. The systematic uncertainties in the performance of the dark shower tagger should then not affect the background estimates, only the signal efficiency. We have substantially rewritten the text to make this point clear.
Comment 9: Last paragraph of the conclusions: unsupervised methods are not the only anomaly detection procedures that have been proposed - there are a variety of semi-supervised methods that may be (more) effective. See e.g. https://iml-wg.github.io/HEPML-LivingReview/.
Reply 9: We now also mention semi-supervised methods.
Comment 10: Delphes: what detector setup are you using?
Reply 10: We have added this information (when we perform a detector simulation, we use the ATLAS card, in particular for the mono-jet analysis).
Report 4:
Comment 1: In section 2, the authors describe the model and how it can lead to semi-invisible jets. While the diagram is nice, showing how the Z' decays, it would be very helpful to explicitly show the decay modes of the ρ+,ρ0. This would help explain why the ρ0 is the only dark meson decaying on collider time scales and setting the R_inv = 0.75. For instance in [1809.10184], the ρ+ decays promptly to q q' when mρ < 2 mπ.
Reply 1: Indeed, r_inv is model-dependent and it is important to consider the effect of varying it. For our benchmark choice r_inv = 0.75, we rely on the specific model from arXiv:1907.04346, where all SM quarks have the same U(1)' charge and hence the charged dark rho mesons cannot decay into a quark-antiquark pair. We feel that the present paper is not the right place for discussing these details and have therefore decided to give only a brief summary and refer the reader to earlier publications.
Comment 2: In Figure 3 the CNN and LoLa change which is better for tops versus semi-visible. Do the authors have any insights into this?
Reply 2: We have not investigated this effect further.
Comment 3: Table 2 shows the results for training and testing on different meson masses. However, they never show results for testing on a lighter mass than was trained on, is there a reason for this.
Reply 3: The effects when testing on lighter masses than the one used for training is completely analogous to when testing on heavier masses.
Comment 4: Using a parameterized network would greatly help over the mixed samples. These allow the network to generalize between samples better. The authors mention this but do not cite the paper showing this [https://arxiv.org/abs/1601.07913 for hep]
Reply 4: We have added this reference, which was indeed missing in our discussion.
Comment 5: On page 12, the number 3.84 seems to come out of nowhere. Rephrasing such that they are looking for the 95% CL exclusion, which is when q_mu = 3.84 would help the reader.
Reply 5: See above (report 3, reply 7).
Comment 6: On page 13, there needs to be more explanation about why systematics can now safely be neglected, especially since the previous section was about how training on the wrong parameters (even on mixed samples) leads to a reduction in performance.
Reply 6: See above (report 3, reply 8).
Comment 7: They should also cite the few other hep-ph papers utilizing graph neural networks such as https://arxiv.org/abs/1807.09088, (https://iml-wg.github.io/HEPML-LivingReview)
Reply 7: We have added a number of relevant reference.
Report 1:
Comment 1: The high level variables referred to at the bottom of p. 2 are conventionally referred to collectively as N-subjettiness, not subjettiness.
Reply 1: Changed as suggested.
Comment 2: Given that baryons would be expected to have a mass ∼Λd, the same as the assumption made for the mesons in the text, better justification for ignoring baryons in the phenomenological study should be given. Since an SU(3) gauge group is assumed, I would expect a combinatoric suppression of ∼1/9 in collider production, as in the SM. The authors should comment on whether this is enough to ignore the baryons for their purposes or if additional suppression must be provided.
Reply 2: If baryons are produced in the dark shower their only effect would be to slightly increase the fraction of invisible particles r_inv. Since we consider different values of r_inv, our results therefore do not rely on any specific assumption regarding the production of baryons in the dark shower. Nevertheless, the reviewer is correct that our benchmark choice r_inv = 0.75 assumes negligible contribution from baryons (we have now made this assumption explicit in the text). The reason for this assumption is that in the specific case we consider, the masses of dark mesons and dark baryons are determined not only by Λd, but also by the masses of the dark quarks, and hence it is to be expected that the baryons are somewhat heavier than the mesons. Nevertheless, we feel that the present paper is not the right place to discuss these details and have therefore decided to keep the discussion short. We also note that it is presently not possible to simulate the production of dark baryons with the Hidden Valley module of Pythia.
Comment 3: Since no hyperparameter optimization is performed for the networks going from top to semi-visible jet tagging, can the authors exclude the possibility that the performance differences in the networks would be reduced for a difference choice of hyperparameters?
Reply 3: We thank the reviewer for raising this point. We have tried a number of different ways of modifying the structure and hyperparameters of the CNN and the LoLa network, finding no improvement over the performance shown in our plots. We have added a corresponding comment.
Comment 4: While the DGCNN showed the best performance for a given choice of parameters when trained on those parameters, when ultimately trained with mixed samples (a much more realistic scenario) the performance is degraded by a factor of a few. Have the authors checked that the other architectures degrade similarly when training on mixed samples? If so, this should be stated; if not, it should be done. Perhaps the other architectures are more robust to model variation.
Reply 4: The performance of all three neural networks is substantially degraded when training on the wrong meson mass and slightly degraded when training on mixed samples. While the effect is largest for the DGCNN (reflecting the fact that this architecture is able to learn more details of the semi-visible jets), it still easily outperforms the other two network types for all combinations that we have considered. We have added footnote 3 to clarify this point.
Comment 5: I want to second the other referee's concern about ignoring systematic uncertainties induced by the tagger. The uncertainty in the signal efficiency in real world events for a tagger trained on simulated signal where no calibration region exists can be significant and should be discussed.
Reply 5: See above (report 3, reply 8).
Report 5:
Comment 1: Learnings with low level information (particle level) generally suffer from pileups, especially dealing with jets. There are prescriptions to reduce effects from pileups. How much is DGCNN robust under these conventional procedures for pileups removals ?
Reply 1: We thank the reviewer for this interesting question. We have started looking into the effects of pile-up, but the effort required for a conclusive study are beyond the scope of the present work. Nevertheless, we now mention this important direction for future work in the conclusions.
Comment 2: It would be appreciated if authors perform comparison analyses between semi-invisible and hadronic tau. On top of this, W+jets (where W->hadronic tau+neutrino) study would be relevant for mono-jet study.
Reply 2: For the lightest dark meson masses that we consider (m_meson = 5 GeV) and assuming that hadronic taus have not been included in the background sample, the network indeed does not reject hadronic taus as efficiently as QCD jets. However, hadronic taus are in principle easy to distinguish from semi-visible jets, as they have much fewer constituents. Since a simple cut on the number of constituents already removes most hadronic taus, we expect that training the network on a background sample containing hadronic taus will make it possible to efficiently suppress this background.
Comment 3: Authors "observe" performance of various DL techniques on a subject of "semi-invisible jet" (just like HEP-EX.) It would be very welcomed if authors provide some reasons (qualitative analyses) behind these performance dependency on Dark QCD model-parameters.
Reply 3: The discussion of model dependence in section 3.2 is intended precisely for this purpose, i.e. to identify the dependence of the performance of the network on the model parameters. For a more detailed discussion of how the properties of dark showers depend on the underlying model, we refer to the recent paper arXiv:2007.11597.
Comment 4: I guess that the Z' mass dependence in analysis is negligible as authors seem like to fix PT range of jet. For Z' = 2TeV where PT of jet are O(1) TeV, what would be expected performance over top-jet with O(1) TeV PT ?
Reply 4: The reason that we focus on relatively low pT is that most of the dark showers remain invisible. So the transverse momentum distribution of semi-visible jets falls steeply above a few hundred GeV even if the dark quarks are produced with TeV-scale energy. Generating samples of semi-visible jets with very high pT is hence computationally very expensive and also not very promising, given that the signal efficiency will be very low irrespective of the performance of the tagger.

---

## Round 3 · Referee Report · Anonymous · 2021-1-8

Report

Thank you for taking into account my feedback. I am happy to recommend publication of the current manuscript.

---

## Round 3 · Referee Report · Anonymous · 2021-2-2

Report

I am happy with the minor changes that have been made and am happy to recommend publication.

---

## Round 3 · Author Response

We have made a few more changes to the manuscript following the suggestions made by the reviewers.

---

## Editorial Decision

published